# Serum Omentin Levels in Patients with Prostate Cancer and Associations with Sex Steroids and Metabolic Syndrome

**DOI:** 10.3390/jcm9041179

**Published:** 2020-04-20

**Authors:** Artur Borowski, Lucyna Siemińska

**Affiliations:** 1The Silesia Center of Urology Urovita, 41-500 Chorzów, Poland; arturstborowski@gmail.com; 2Department of Pathophysiology and Endocrinology, School of Medicine with the Division of Dentistry, Medical University of Silesia, 41-800 Zabrze, Poland

**Keywords:** prostate cancer, omentin, sex steroids, SHBG, metabolic syndrome

## Abstract

Mechanisms linking obesity and prostate cancer (PC) include increased insulin signaling, persistent inflammation, and altered adipocytokines secretion. Previous studies indicated that omentin may play a potential role in cancerogenesis of different sites, including the prostate. In this study, we focused on the hormonal and metabolic characteristics of men recruited for prostate biopsy. We evaluated serum concentrations of adipocytokines and sex steroids where concentrations are related to the adiposity: omentin, leptin, testosterone, estradiol, and sex hormone-binding globulin (SHBG). Aim: The aim of the study was to assess the concentration of serum omentin in men with PC. We also investigated relationships between omentin, leptin, sex steroids, SHBG, age, and metabolic syndrome (MS). Methods: Our study was conducted on 72 patients with PC and 65 men with benign prostate hyperplasia (BPH). Both groups were compared for body mass index. Results: Comparing men with PC to subjects with BPH there were significantly higher serum concentrations of omentin, estradiol, and prostate specific antigen (PSA) in the former. Estradiol/testosterone ratio, which is a marker of testosterone to estradiol conversion, was also significantly higher in the PC group. MS was diagnosed in 47 men with PC and in 30 men with BPH, the prevalence was significantly higher in the PC group. When the subjects with PC were subdivided into two subgroups, the serum omentin did not differ between those with MS and without MS. In the overall sample serum, omentin was positively associated with age, SHBG, and leptin. A positive correlation was also found between omentin and estradiol/testosterone ratio, and negatively with testosterone/SHBG ratio. Positive correlations were noted between age and SHBG, PSA and estradiol/testosterone ratio. In our study, a drop of total testosterone and testosterone/SHBG ratio, due to age, was also demonstrated. Conclusions: In patients with prostate cancer, serum omentin may be a diagnostic indicator. Omentin levels do not correlate with estradiol or testosterone concentrations but they are related to the testosterone/SHBG ratio. Omentin is not associated with an increased likelihood of having metabolic syndrome in men with prostate cancer.

## 1. Introduction

Epidemiological data indicate a progressive increase in the incidence of prostate cancer (PC) in all age groups. In the United States, in 2019, PC was the most prevalent cancer among males [1]. Known risk factors include age, family history/genetic factors (HPC-hereditary PC), dietary habits, and reduced physical activity. The role of obesity, metabolic syndrome and adipocytokines in the etiology of PC have been intensively studied [2,3]. PC develops from high-grade prostatic intraepithelial neoplasia, however, critical for the initiation of cancerogenesis and tumor progression are different mechanisms that take place in the periprostatic adipose tissue which surrounds and penetrates the prostatic gland, especially in obese men. It is considered that adipocytes surrounding the tumor may play a role as an energy source for growing the tumor and can become a source of lipids transferring into cancer cells. Accumulation of lipids in cancer cells is the characteristic feature of prostatic cancerogenesis [4]. Adipocytes and other cells of adipose tissue secrete adipokines and cytokines including leptin, adiponectin, resistin, visfatin, omentin, and IL-6. It is likely that bioactive proteins at both systemic and local levels, contribute to oxidative stress, DNA damage and through various mechanisms such as modulation of lipolysis/lipogenesis/beta-oxidation, interactions with hormone-dependent pathways, promoting proliferation and migration of cancer cells and inducing of differentiation of preadipocytes into fibroblasts, may exert effects on cancerogenesis [3]. It was found that PC cells alter adipocytokines secretion from periprostatic adipose tissue, thus interactions between adipocytes, stromal and cancer cells are mutual.

The role of leptin and adiponectin in prostate cancerogenesis is well documented [2,5,6,7]. Leptin is a lipid-related hormone and its serum concentration is positively associated with body fat storage. It is believed that leptin secreted abundantly by excessive adipose tissue into the systemic circulation and locally released from adipose tissue overgrowing nearby the prostate gland, shows a pro-carcinogenic effect. In vivo studies revealed that leptin has a direct effect on the prostate gland leading to the proliferation of epithelial prostatic cells [6,8]. Leptin stimulates migration and invasion of PC cells, as well as promotes neoangiogenesis through releasing vascular endothelial growth factor (VEGF). Moreover, leptin suppresses apoptosis of cancer cells and increases chronic inflammation. It was shown that PC cells have high expression of leptin receptors [6]. A higher incidence of progressive and invasive PC with poor prognosis was reported in obese men with hyperleptinemia [5]. In our previous study, we were unable to find any differences in leptin levels between PC and benign prostate hyperplasia (BPH) men, however, we found the higher leptin-to-adiponectin ratio in individuals with PC and we demonstrated connections between serum leptin and aggressiveness of PC [9]. The pathophysiological role of other adipocytokines is not yet fully understood. Recently discovered adipocytokine is omentin, also known as intelectin-1. This protein plays an anti-inflammatory, anti-oxidative and anti-diabetic role [10] but the potential role in cancerogenesis remains uncertain. Elevated concentrations of omentin have been described in gastric and colon cancers [11,12]. Several studies have found increased levels of omentin in PC but the underlying mechanism is not fully understood [13,14,15,16]. In this study, we investigated whether serum omentin levels might be a marker for PC.

Cancer of the prostate gland is still considered to be androgen-induced and patients with PC are always treated with androgen-deprivation therapy. However, most prospective and retrospective studies assessing the association of PC with endogenous testosterone, dihydrotestosterone or free testosterone do not show any relationships [17]. It is shown that low levels of endogenous testosterone do not protect against PC and testosterone replacement therapy does not increase the risk [18]. Moreover, there is evidence that low serum testosterone concentrations are associated with worse prognosis and with higher tumor aggressiveness [19]. The prostate is a tissue controlled by estrogens, androgens and sex hormone-binding globulin (SHBG). It is considered that long-term estrogen stimulation and low testosterone concentration, especially in the condition of obesity-induced inflammation, may be a key factor in the initiation of prostate cancerogenesis [20]. Since the secretion of most adipocytokines is fat mass-dependent and androgens and estrogens might regulate fat deposition, we speculate that sex hormones may influence the expression of omentin in adipose tissue and may affect the serum omentin concentration. Knowing the mechanisms of PC development can provide a basis for establishing possible strategies for cancer prevention.

The aim of the present study was to determine concentrations of serum omentin in men with PC, and to compare them with concentrations in BPH. The relationship of omentin to metabolic syndrome, age, serum leptin, sex steroids, SHBG was also investigated.

## 2. Material and Methods

In total, 137 patients between 42 and 85 years of age, were prospectively included in the study. Men were referred to the Urology Center for diagnostic evaluation due to increased prostate specific antigen (PSA) or because of incorrect results in a digital rectal exam or as a result of an invalid examination in ultrasonography or MR. Exclusion criteria were: BMI <20 or BMI >38, heart and liver failures, chronic kidney disease and diagnosis of cancer other than PC. In all patients, a transrectal ultrasound biopsy was conducted. After the biopsy, the subjects were divided into a biopsy-proven PC group (included 72 patients) and a biopsy-negative benign prostate hyperplasia (BPH) group (65 patients). Both groups were compared for body mass index (BMI), mean BMI in the cancer group was 27.90, and 27.23 in the benign hyperplasia group. No patient had distant metastases at the time of diagnosis. Patients with PC were subdivided into 3 groups by Gleason score: well differentiated subgroup (Gleason score ≤6), moderately differentiated subgroup (Gleason 7) and poorly differentiated subgroup (Gleason ≥8). Anthropometric measurements of height (cm), weight (kg), and waist circumference (cm) were done in all subjects and BMI was calculated. For the analysis of serum concentrations of adipocytokines and sex steroids, fasting venous blood samples were collected from each participant in the morning, in the PC group before surgery. The serum was obtained by centrifugation at 3000× *g* for 10 min and then stored at −70 ℃ until assays were performed. Blood pressure was measured at rest. In all men, the metabolic profile was assessed (fasting glucose concentrations and lipid profile: total cholesterol CHOL, high-density lipoprotein cholesterol HDL-C, triacylglycerol TG) using a standard enzymatic method. The presence of metabolic syndrome (MS) was assessed according to the International Diabetes Federation definition (IFD), 2006. We used the following IFD cut-off limits: 1. Fasting glucose ≥100 mg/dL; 2. TG >150 mg/dL; 3. HDL-C <40 mg/dL; 4. Waist circumference ≥94 cm; 5. Blood pressure >130/85 mmHg. MS was diagnosed when central obesity defined as waist circumference ≥94 cm plus any two of four other factors were present. Central obesity was always assumed if BMI was greater than 30 kg/m^2^.

Serum omentin, leptin, testosterone, estradiol, SHBG, and insulin concentrations were assessed by ELISA methods using commercial assays (Omentin-1 Human ELISA-BioVendor, Leptin Human ELISA-BioVendor, Insulin Human ELISA-BioVendor, Estradiol ELISA-DiaMetra, Testosterone ELISA-DiaMetra, SHBG ELISA-DiaMetra). Bioactive testosterone and bioactive estradiol were calculated from their total levels and SHBG concentration and presented as testosterone/SHBG ratio and estradiol/SHBG ratio. Testosterone is converted to estradiol in peripheral tissues, therefore, we also calculated the estradiol/testosterone ratio. Homeostatic Model Assessment of Insulin resistance (HOMA-I) was calculated with the formula: HOMA-I = fasting serum glucose concentration (mg/dL)/18.1 × fasting serum insulin concentration (uIU/mL)/22.5. Serum PSA concentrations were measured using an immunochemical method (ECLIA). Intra and extra assay errors did not exceed 10%. Concentrations of analyzed parameters were presented as means ± SD and medians. For comparisons between the groups, we used the *t*-test for normal distribution or the Mann-Whitney U test for distributions other than normal. Correlations between variables were estimated by calculating the correlation coefficient R by Spearman’s method.

The prevalence of metabolic syndrome was compared using the chi-squared test. Multiple regression analysis was performed to detect associations between the serum omentin concentration and the analyzed parameters. In the model, omentin was the dependent variable and leptin, SHBG, age, estradiol/testosterone ratio, testosterone/SHBG and the presence of PC were independent variables. Statistical analyses were performed using STATISTICA 12.0, (StatSoft Inc, Tulsa, OK, USA), assuming the levels *p* < 0.05 as statistically significant. All subjects who participated in the study provided informed consent to allow an analysis of data for research purposes and all subjects gave the agreement in the written form. The study was approved by the local Ethical Review (the study protocol of Silesian Medical University KNW/0022/KB1/9/13).

## 3. Results

One hundred and thirty-seven subjects were enrolled in the study. Group I consisted of 72 PC men, Group II consisted of 65 BPH men. Groups were compared for BMI. We evaluated the levels of omentin, leptin, sex steroids, SHBG, and metabolic parameters. The results are given in Table 1.

Comparing men with PC to subjects with BPH, there were significantly higher serum concentrations of omentin, estradiol, and PSA in the former. Estradiol/testosterone ratio which is a marker of testosterone to estradiol conversion was also significantly higher in the PC group. No significant differences were found between serum testosterone, SHBG, testosterone/SHBG ratio, estradiol/SHBG ratio and leptin in PC and BPH groups. Individuals with PC were older than the men with BPH, however, when subjects were matched by age, the differences were still significant. 

MS was diagnosed in 47 men with PC and in 30 men with BPH, and the prevalence was significantly higher in the PC group (*p* < 0.05). When we compared men with PC and MS and men with BPH and MS, the serum omentin concentrations were significantly higher in the PC men (Table 2, Figure 1). However, when the male subjects with PC were subdivided into 2 subgroups, the serum omentin did not differ between those with MS and without MS (Table 2, Figure 2). As expected, we observed significantly higher HOMA-I and leptin levels in PC patients with MS compared to the non-metabolic syndrome individuals with PC, the same observations were made for patients with BPH.

Spearman correlation was used to evaluate the linear correlations between analyzed variables in all subjects. The results are given in Table 3. Serum omentin was positively associated with age, leptin, and SHBG. A positive correlation was also found between omentin and estradiol/testosterone ratio, and negative with testosterone/SHBG ratio. In both groups analyzed together, positive correlations were noted between age and omentin, SHBG, PSA and estradiol/testosterone ratio. In our study, a drop of total testosterone and testosterone/SHBG ratio, due to age, was also demonstrated. Leptin positively correlated with BMI, HOMA-I, omentin, estradiol/SHBG ratio and estradiol/testosterone ratio, a negative correlation was found with testosterone. Testosterone positively correlated with estradiol and SHBG. In the PC group, we did not find a significant relationship between serum omentin and the Gleason score.

Variables that were found to be correlated with serum omentin levels in univariate analyses were used in the multiple regression analysis (Table 4). In the analysis where omentin was the dependent variable, the presence of PC and serum both leptin and SHBG remained significantly associated with the increased levels of omentin.

## 4. Discussion

In the last years, significant links of omentin with different malignancies, especially colorectal cancer, was demonstrated [12,13]. Higher concentrations of omentin were also reported in mesothelioma [21], in gastric cancer [11], and pancreatic cancer [22] and in most of those studies, increased omentin levels were independent of BMI, glucose, and lipid parameters. Increased concentrations of omentin in PC cases were described previously by Zhou et al. [14] and in the Turkey population by Uyeturk et al. [15]. Higher levels of serum omentin in PC were also observed in the Polish patients [16]. The case-control study conducted by Fryczkowski et al. on 40 patients with PC and 40 patients with BPH, reported significantly elevated concentrations of omentin in patients with cancer. The authors propose that this adipocytokine could be a noninvasive biomarker of PC [16]. Indeed, the decision on therapy is currently based mainly on PSA serum assessment with frequent false-positive results leading to unnecessary overtreatment. The finding of other simple biomarkers isolated from the blood would avoid unnecessary treatment of patients with PC. Our findings revealed that serum concentrations of omentin were significantly higher in PC subjects than in men with prostate hypertrophy and this phenomenon was independent of BMI and metabolic syndrome. We did not observe any relationships between the Gleason score and omentin levels and this observation correlates with previous findings [14,15]. Additionally, we demonstrated positive correlations between omentin and leptin. Because leptin is a procarcinogenic adipocytokine [5,6], increased levels of omentin are probably linked with the oncogenic effect.

The views of effects of omentin on carcinogenesis are inconclusive because clinical and laboratory results show various, often paradoxically opposite mechanisms of omentin action. The detailed role of the omentin in cancerogenesis is not well characterized and the reasons for increased omentin concentrations in cancers are unknown. Different results of in vitro and in vivo studies uncovered that omentin may act as the tumor suppressor. It has been shown that omentin inhibited the proliferation and promoted apoptosis of human hepatocellular carcinoma HepG2 and HuH-7 cells, and as Zhang et al. uncovered, this effect was obtained by activation of the JNK signaling pathway and p53 up-regulation [23]. Mogal et al. have demonstrated that omentin suppresses prostate cell growth, proliferation and viability via inhibiting the effect of Nkx3.1 [24]. Furthermore, Ji et al. found that omentin inhibited the proliferation and promoted apoptosis of colon cancer stem cells [25]. Based on these results, omentin can be considered as an anti-cancer factor and its elevated concentrations could be a compensatory mechanism. However, data regarding the effects of omentin on carcinoma cell proliferation reported by other researchers are contradictory and they show omentin like a factor promoting cancer growth. Yan et al. found that omentin directly induced the proliferation of colon cancer cells [26]. Moreover, the results obtained by Zhang et al. showed that colorectal carcinoma cells express and secret omentin into local tissues. Authors speculated that this adipocytokine serves cancerogenic effects through autocrine, paracrine, and endocrine mechanisms [27]. Procancerogenic effect of omentin may depend on enhancing glucose uptake [28]. Another potentially plausible mechanism includes the stimulating effect of omentin on cell proliferation by triggering PI3K/Akt signaling [29]. As demonstrated in recent years, the activated PI3K/Akt/mTOR pathway is one of the crucial players in prostate cancerogenesis [30]. Omentin differs from other adipocytokines because it is produced by the stromal-vascular fraction of visceral adipose tissue. It was shown that this protein, through PI3K/Akt signaling pathway, promotes the proliferation of mesenchymal stem cells and inhibits their apoptosis [31]. Mesenchymal stem cells, together with adipocytes, macrophages, and endothelial cells, form the tumor microenvironment [3]. It is a heterogeneous population of multipotent stromal cells that functionally are characterized by the capability to differentiate into different cells like adipocytes, fibroblasts, and smooth muscle cells. They may play a role in cancerogenesis via multiple pathways, including secretion of proangiogenic substanties (e.g., VEGF) and upregulation of the fibroblast growth factor, which is essential for growth and neovascularization. They can also migrate to local and distant areas of the tumor and prostate gland [32]. The experiment conducted in vivo and in vitro by Chang et al. demonstrated that mesenchymal stem cells had an effect on the transformation of androgen-dependent human prostate carcinoma cells in an androgen-independent manner. This phenomenon is prognostically unfavorable [33]. The regulation of omentin secretion is not fully understood. In vitro and in vivo studies revealed decreased production of omentin under glucose or insulin infusion [34]. Most articles published so far showed that omentin concentrations were down-regulated in insulin-resistant states such as obesity, polycystic ovary syndrome, gestational diabetes mellitus, and type 2 diabetes [34,35,36]. Some studies demonstrated that serum omentin levels were elevated in women when compared with males [36]. Negative correlations were found between omentin and androgen levels in polycystic ovary syndrome (PCOS) women, independent of body mass [37]. Moreover, higher concentrations of omentin were observed in rats which received estrogen therapy [38]. In the current study, we found that omentin did not correlate with estradiol or testosterone concentrations. However, we observed negative associations between bioactive testosterone, calculated as testosterone/SHBG ratio and omentin, and positive relations between SHBG and omentin. In the blood, sex hormones are transported to target cells by SHBG, therefore, this carrier protein determines their biological activity. SHBG is synthesized in the liver under androgens and estrogens control, so relationships between SHBG and sex steroids are bidirectional. SHBG levels are influenced by insulin, nutritional and metabolic factors. One of the strongest regulators of SHBG is hepatocyte nuclear factor 4 alpha (HNF 4-α) [39]. It is a member of the nuclear receptor superfamily which regulates many genes involved in glucose, cholesterol, and fatty acid metabolism. It was demonstrated as high HNF 4-α expression in human liver, intestine, pancreas [40], and was recently demonstrated in epithelial cells of both normal prostate as well as PC [41]. In prostate cells, HNF 4-α is linked to the androgen receptors but the nature of signaling remains unclear. Increased expression of HNF 4-α in different carcinomas including ovarian, colorectal, lung, neuroblastoma was shown [41], however, further studies are needed to determine the role of HNF 4-α in prostate cancerogenesis. Hyperglycemia and a high carbohydrate diet which decrease HNF 4-α transcript levels, simultaneously reduce SHBG expression and serum SHBG concentrations [39]. In turn, those factors which increase HNF 4-α expression like adiponectin and thyroid hormones, lead to higher SHBG production [42]. Recently, Li et al. demonstrated that omentin can increase HNF 4-α expression in gastric cancer tissues [43], therefore, it can be assumed that high omentin level, through HNF 4-α, can be associated with high SHBG level. In the present study, omentin correlated positively with SHBG, but the precise character of this relationship is not known. Since HNF 4-α, SHBG and sex steroids are associated with intracellular lipid incorporation and release, it seems likely that omentin may be a link between lipogenesis/lipolysis balance, however, in order to explore specific mechanisms connecting omentin and lipids, specific future studies are needed. It was shown that PC cells and tumor-surrounding adipocytes exhibit enhanced release of adipocyte-derived lipids as well as increasing rates of de novo lipids synthesis; therefore, perturbed lipid metabolism is a characteristic feature of prostatic cancerogenesis [4].

Most studies demonstrated that in men, SHBG concentration increases with age [44,45]. As in previous studies, our analysis found the strong relationship between SHBG and age. It should be noted that not only SHBG positively correlated with age, but also similar connections were observed between omentin and age, and this is consistent with the data described by other researchers [46]. Because aging is associated with the accumulation of visceral fat, it seems that redistribution of lipids from the subcutaneous to the abdominal compartment explains the increasing levels of omentin and SHBG. Lower physical activity, chronic excess of calories, and reduced basal metabolic rate are the reasons for age-related abdominal fat mass expansion. Additionally, declining testosterone and growing of both SHBG and estrogens as men get older, all contribute to the accumulation of visceral adiposity. In our study, we found an age-related decline in total testosterone and testosterone/SHBG ratio. We also showed that conversion of testosterone to estradiol, presented as estradiol/testosterone ratio, also increased with age. This phenomenon can be attributed to increased activity of aromatase due to the cumulation of fat mass. Our data were consistent with the results of previous publications [47].

Most researchers described that omentin was decreased in both metabolic syndrome and diabetes [10,34,35], however, we did not observe differences in circulating omentin concentrations between individuals with and without metabolic syndrome, in both groups. What is worth noting, men with PC and metabolic syndrome had significantly higher serum omentin concentrations than men with BPH and metabolic syndrome. The data regarding relationships between omentin and metabolic risk factors are not entirely clear. It is assumed that omentin is a beneficial adipocytokine which exerts favorable metabolic effects [10]. The results of an in vitro study demonstrated that omentin enhances insulin sensitivity, reduces glucose levels, and regulates lipid metabolism. However, in the study conducted on 684 middle-aged population, men with higher omentin tertile had a more unfavorable metabolic profile and higher systolic blood pressure than men in the lower omentin tertile [48]. Recently, it was shown that in subjects with diabetes, higher omentin concentrations were associated with increased risk for cardiovascular events [49], moreover elevated omentin levels were observed in nonalcoholic fatty liver disease and in inflammatory states [50]. Similar unclear associations relate SHBG to health. Decreased SHBG levels are usually accompanied by abnormal metabolic status and predict the development of diabetes mellitus [51]. Surprisingly, epidemiological studies showed that not low but high serum SHBG was associated with increased all-cause mortality in men with type 2 diabetes [52] and was connected with the increased risk of cardiovascular diseases [45]. The reasons for those discrepancies are completely unclear and previous studies did not explain the nature of this phenomenon. In our study, when we compared PC and BPH groups, there were no differences in serum testosterone concentrations. However, we revealed higher concentrations of estradiol in PC patients when compared to BHP men, and it is consistent with the results of some previous publications [53]. Some authors speculated that higher concentrations of estrogens in African Americans compared to Caucasian Americans, as well as higher estradiol/testosterone ratio, may contribute to their increased risk of PC [54]. Estrogens are produced in adipose tissue from androgens through enzyme aromatase and their role in prostate cancerogenesis appears to be complex [55]. It is considered that estrogens through ER-α stimulates cancerogenesis, while ER-β plays a more protective role, mainly inhibits tumor progression and causes apoptosis. Elevated estrogen concentrations stimulate prostate cancerogenesis, however, when concentrations are even more increased, an inhibitory effect appears. In vivo studies revealed that prostatic intraepithelial neoplasia increased 3-fold in rats which received simultaneously estradiol and testosterone, compared to animals that received only testosterone [55]. The serum sex steroid concentrations might not reflect their actual levels in adipose tissue, however, in our study, serum estradiol did have a positive correlation with testosterone, indicating that estradiol was testosterone dependent. Higher omentin and estradiol concentrations in our patients with PC suggest that both hormones have a role in PC development. In our previous study, we explored the ratio of leptin to adiponectin in PC cases and we revealed that the balance was disturbed [9]. Other researchers considered that interplay between high estradiol, leptin, and insulin may determine androgen-dependent PC development [53,55]. An in vivo study conducted by Alves-Pereira et al. revealed that leptin increased the proliferation of prostate epithelial cells and the expression of aromatase and estrogen receptors [8].

The limitation of the study was that serum sex steroids and adipocytokines were measured at one time point and may not be representative of tissue levels. Moreover, we measured only BMI and waist circumferences without assessing fat mass and its distribution.

## 5. Conclusions

In patients with prostate cancer, serum omentin may be a diagnostic indicator. Omentin levels do not correlate with estradiol or testosterone concentrations but they are related to the testosterone/SHBG ratio. Omentin is not associated with increased likelihood of having metabolic syndrome in men with prostate cancer.

## Figures and Tables

**Figure 1 jcm-09-01179-f001:**
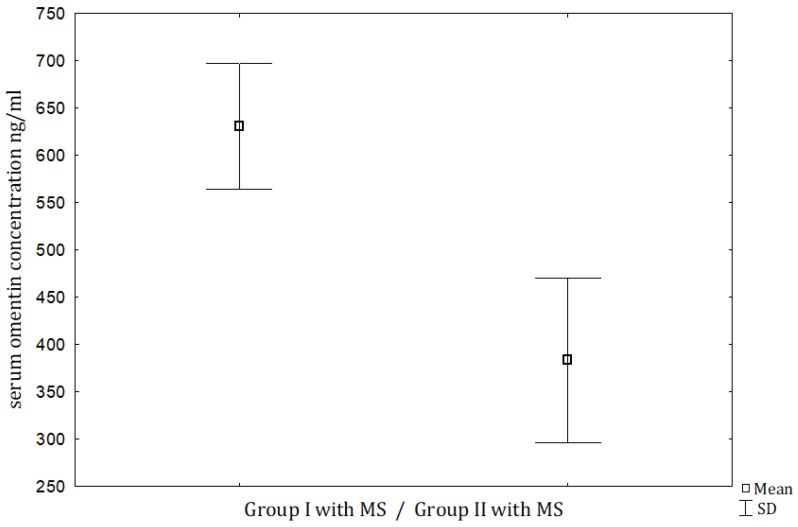
Comparison of serum omentin between men with PC (Group 1) and metabolic syndrome versus men with BPH (Group 2) and metabolic syndrome. *p* < 0.001. PC, prostate cancer; BPH, benign prostate hyperplasia; MS, metabolic syndrome; SD, standard deviation.

**Figure 2 jcm-09-01179-f002:**
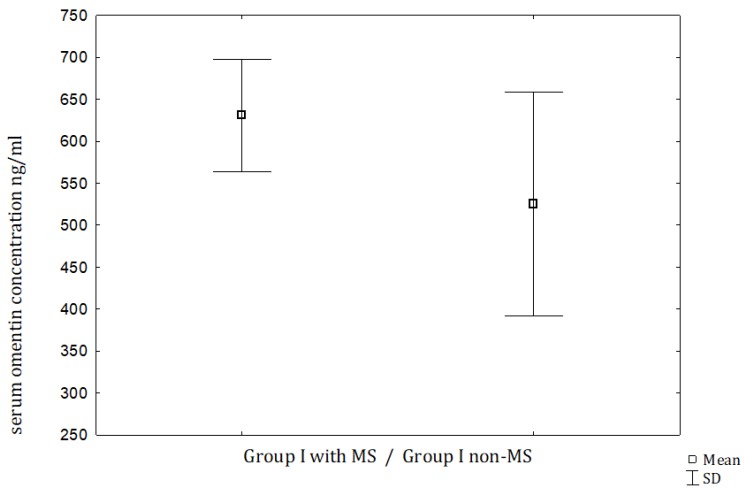
Comparison of serum omentin between men with PC and metabolic syndrome versus men with PC without metabolic syndrome. *p* = NS, not statistically significant.

**Table 1 jcm-09-01179-t001:** The clinical, hormonal and metabolic parameters of studied men and comparison between the groups (Group I: PC (prostate cancer), Group II: BPH (benign prostate hyperplasia)).

	PC (Group I), *n* = 72 Mean ± SD (Median)	BPH (Group II), *n* = 65 Mean ± SD (Median)	*p*
Age	67.08 ± 8.36 (67.0)	61.43 ± 10.17 (60.0)	<0.001
BMI (kg/m^2^)	27.90 ± 3.29 (28.0)	27.23 ± 4.00 (26.57)	Ns
Waist circumference (cm)	102.38 ± 9.93 (102.0)	101.72 ± 9.33 (101.0)	Ns
Fasting glucose (mg/dL)	104.76 ± 28.20 (103.0)	108.33 ± 33.57 (98.0)	Ns
CHOL (mg/dL)	185.11 ± 45.99 (182.5)	187.41 ± 45.41 (187.0)	Ns
HDL-C (mg/dL)	47.26 ± 14.03 (45.0)	50.75 ± 22.42 (48.0)	Ns
TG (mg/dL)	124.42 ± 49.74 (113.0)	138.15 ± 92.03 (110.0)	Ns
HOMA-I	3.88 ± 4.27 (3.22)	5.28 ± 8.20 (2.87)	Ns
Omentin (ng/mL)	594.29 ± 266.85 (565.3)	379.92 ± 168.05 (360.9)	<0.001
Leptin (ng/mL)	10.04 ± 8.18 (8.25	9.02 ± 7.23 (7.87)	Ns
Testosterone (ng/mL)	3.64 ± 2.79 (3.36)	3.49 ± 1.37 (3.43)	Ns
Estradiol (pg/mL)	24.85 ± 20.79 (21.1)	19.27 ± 9.13 (18.4)	<0.01
SHBG (nmol/L)	30.18 ± 13.09 (27.8)	26.27 ± 15.14 (24.6)	Ns
Testosterone/SHBG ratio	0.20 ± 0.59 (0.12)	0.17 ± 0.10 (0.15)	Ns
Estradiol/SHBG ratio	1.37 ± 2.45 (0.72)	1.06 ± 0.84 (0.88)	Ns
Estradiol/Testosterone ratio	15.63 ± 27.96 (7.87)	7.36 ± 8.16 (5.56)	<0.01
PSA (ng/mL)	33.85 ± 91.09 (7.8)	3.80 ± 5.43 (1.51)	<0.001

PC, prostate cancer; BPH, benign prostate hyperplasia; BMI, body mass index;, Ns, not statistically significant; CHOL, cholesterol; HDL-C, high density lipoprotein cholesterol; TG, triacylglycerol; HOMA-I, homeostatic model assessment of insulin resistance; SHBG, sex hormone-binding globulin; PSA, prostate specific antigen.

**Table 2 jcm-09-01179-t002:** The clinical, hormonal, and metabolic characteristics in PC and BPH men stratified by the presence/absence of metabolic syndrome (MS).

	PC (Group I), *n* = 72		BPH (Group II), *n* = 65		Differences between Group			
	Group I MS, *n* = 47 Mean ± SD (Median)	Group I non-MS, *n* = 25 Mean ± SD (Median)	Group II MS, *n* = 30 Mean ± SD (Median)	Group II non-MS, *n* = 35 Mean ± SD (Median)	Group I MS vs. Group I non-MS (*p*)	Group II MS vs. Group II non-MS (*p*)	Group I MS vs. Group II MS (*p*)	Group I non-MS vs. Group II non-MS (*p*)
Age	66.89 ± 8.12 (68.0)	67.44 ±8.97 (66.0)	63.90 ± 10.184(62.0)	59.31 ± 9.82 (57.0)	NS	NS	NS	<0.01
BMI (kg/m^2^)	29.11 ± 2.83 (28.40)	25.61 ± 2.84 (25.84)	28.79 ± 4.15 (27.93)	25.95 ± 3.43 (25.66)	<0.001	<0.01	NS	NS
Waist circumference (cm)	106.29 ± 8.07 (105.0)	95.04 ± 8.96 (94.0)	106.56 ± 7.47 (106.5)	97.57 ± 8.83 (98.0)	<0.001	<0.001	NS	NS
Fasting glucose (mg/dL)	110.65 ± 31.98 (106)	93.68 ± 13.99 (96.0)	126.53 ± 41.60 (113.5)	92.74 ± 10.15 (91.0)	<0.001	<0.001	NS	NS
CHOL (mg/dL)	187.40 ± 45.85 (181.0)	180.80 ± 46.89 (186.0)	195.06 ± 49.64 (194.5)	180.85 ± 41.03 (184.0)	NS	NS	NS	NS
HDL-C (mg/dL)	46.40 ± 12.65 (44.0)	48.88 ± 16.46 (50.0)	43.16 ± 10.21 (45.5)	56.43 ± 28.16 (49.0)	NS	<0.05	NS	NS
TG (mg/dL)	139.59 ± 53.57 (123.0)	95.88 ± 22.76 (92.0)	157.06 ± 104.8 (122.0)	121.94 ± 77.34 (97.0)	<0.001	NS	NS	NS
HOMA-I	4.76 ± 4.97 (3.56)	2.22 ± 1.47 (2.01)	8.25 ± 10.92 (4.80)	2.41 ± 1.37 (1.88)	<0.05	<0.01	NS	NS
Omentin (ng/mL)	630.81 ± 227.06 (576.5)	525.65 ± 323.0 (453.4)	383.22 ± 202.1 (383.0)	377.48 ± 141.11 (341.1)	NS	NS	<0.001	<0.05
Leptin (ng/mL)	12.20 ± 8.18 (10.09)	5.96 ± 6.58 (4.58)	11.54 ± 8.34 (10.37)	6.86 ± 5.35 (5.41)	<0.01	<0.01	NS	NS
Testosterone (ng/mL)	3.31 ± 2.85 (2.91)	4.25 ± 2.62 (3.51)	3.20 ± 1.33 (2.97)	3.74 ± 1.37 (3.58)	NS	NS	NS	NS
Estradiol (pg/mL)	23.59 ± 20.43 (18.2)	27.22 ± 21.68 (24.6)	21.30 ± 9.34 (18.85)	17.53 ± 8.71 (17.1)	NS	NS	NS	<0.05
SHBG (nmol/L)	30.08 ± 11.96 (27.0)	30.38 ± 15.26 (30.6)	25.37 ± 13.86 (23.35)	27.04 ± 16.33 (24.6)	NS	NS	NS	NS
Testosterone/SHBG ratio	0.12 ± 0.12 (0.10)	0.35 ± 0.98 (0.14)	0.16 ± 0.09 (0.15)	0.18 ± 0.11 (0.16)	NS	NS	NS	NS
Estradiol/SHBG ratio	1.47 ± 2.87 (0.68)	1.19 ± 1.43 (0.81)	1.13 ± 0.92 (0.97)	0.96 ± 0.77 (0.73)	NS	NS	NS	NS
Estradiol/Testosterone ratio	15.98 ± 18.97 (8.37)	14.99 ± 40.27 (6.76)	8.25 ± 7.39 (6.67)	6.58 ± 8.82 (5.12)	NS	NS	<0.01	NS
PSA (ng/mL)	39.66 ± 107.03 (7.8)	22.48 ± 46.24 (7.41)	3.34 ± 3.38 (1.55)	4.16 ± 6.63 (1.27)	NS	NS	<0.01	<0.01

**Table 3 jcm-09-01179-t003:** Spearman’s coefficients of the relationships among analyzed variables in all subjects.

	Omentin		Leptin		Testosterone		Age	
	*R*	*p*	*R*	*p*	*R*	*p*	*R*	*p*
Age	0.38	<0.001			−0.17	<0.05		
BMI			0.56	<0.001				
HOMA−I			0.44	<0.001	−0.22	<0.01		
Omentin			0.34	<0.001			0.38	<0.001
Leptin	0.34	<0.001			−0.20	<0.01		
Testosterone			−0.20	<0.01			−0.17	<0.05
Estradiol					0.28	<0.001		
SHBG	0.34	<0.001			0.21	<0.01	0.29	<0.01
Testosterone/SHBG	−0.41	<0.001					−0.41	<0.001
Estradiol/SHBG			0.17	<0.05				
Estradiol/testosterone	0.21	<0.01	0.28	<0.001	−0.62	<0.001	0.27	<0.001
PSA							0.21	<0.01

**Table 4 jcm-09-01179-t004:** Multiple regression analysis for serum omentin. Dependent variable: omentin.

Independent Variable	Regression Coefficient	*p*
Leptin	0.38	<0.001
SHBG	0.16	<0.05
Age	0.11	Ns
Estradiol/testosterone ratio	−0.05	Ns
Testosterone/SHBG ratio	0.05	Ns
PC, yes/no	0.34	<0.001

Ns, not statistically significant.

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
