# Peer review of "Serum Omentin Levels in Patients with Prostate Cancer and Associations with Sex Steroids and Metabolic Syndrome"

_jcm, 2020, doi:10.3390/jcm9041179_

Round 1

Reviewer 1 Report

Serum omentin levels in patients with prostate cancer and associations with sex steroid and metabolic syndrome

Borowski & Sieminska present data on the hormonal and metabolic characteristics of 72 men with prostate cancer, compared to 65 men with benign primary hyperplasia, building on the suggestion that omentin has a role a prostate tumorigenesis. 

Minor points to address

Methods

  • Please provide additional detail/references for the methods used: blood centrifugation (line 81), which ELISA commercial assays (line 91).  Please also clarify HOMA-I.
  • I assume it is simply a feature of the language, but were the prospectively collected patients in this study actively matched for BMI, or was this non-significant difference a happy accident? It isn’t clear to me from the Abstract (groups were compared for BMI, line 20), Methods (groups were compared and line 73) and Results (matched for BMI, line 107) what the case is.

Discussion

In my opinion this section could be streamlined to focus more on the results presented in this report.  There are also several sections I feel could be moved to the introduction, for more general background.

Abstract

In my opinion, the abstract conclusion could be clearer in wrapping up the main stated aim, and suggesting some implications for the authors’ findings in the context of men with PC.  This is suggested in lines 61-62 (Knowing the mechanisms of cancer development can provide a basis for establishing possible strategies for cancer prevention), but not recapitulated or expanded upon in the conclusion.

Major points

Results

Figure 1 as presented here seems redundant; this information is adequately contained in the text.  Perhaps a graphical representation of the more complicated relationship between MS/non-MS PC and BPH and omentin (and others) would be more informative? ( i.e. a subset of the information in Tables 2 and 3).  A visual representation would certainly illustrate the various associations even more clearly than the data tables do alone.

Discussion

The authors describe several potential mechanisms of action of omentin in prostate tumorigenesis (e.g. lines 207-211).  They also speculate on a role for omentin in the tumour microenvironment (lines 219-223), the influence of sex hormones on its expression in adipose tissue (lines 228-231), and its role in PC tumorigenesis with estradiol (lines 312-315).  However, I don’t feel there is sufficient direct data to support these claims in this report, right now.  At the same time however, I realise that the authors probably do have access to tissues from the same patients under study here, to test at least some of these ideas more fully (by IHC or qPCR analysis, for example).  In my opinion, additional data (following up targets they identify  - lines 209-223, 230-231, 312-315) would better support their findings, and improve the report significantly.

Reviewer 2 Report

The authors present interesting results concerning the level of serum omentin in men with PC and its relationship with age, serum leptin, sex steroids and SHBG.

  1. The English language is correct but there are some mistakes with plural which should be verified e.g. Line 64: The relationship of omentin to metabolic syndrome, age, serum leptin, sex steroids, SHBG is also investigated. Probably the past time will be more adequate.
  2. In the method section (line 98) Authors wrote that the “Correlations between variables were estimated by calculating the correlation coefficient R by Pearson`s method or Spearman`s method” however only Spearman results are present in Table 3.
  1. In the Result section (line 129) Authors mentioned that “To assess which factors affect omentin concentration, next we calculated linear correlations between serum levels of omentin and other parameters in the overall subjects. The results are given in Table 3.

It should be commented and explained by the Authors because the strength of the linear relationship between normally distributed variables is assessed by Pearson and  when the relationship between the variables is not linear, it is more appropriate to use the Spearman rank correlation method.

4. There is lack of information about algorithm used for linear regression (In the method section is written that this information is in the result section but there is only information about the results obtained by linear regression).

5. In the Tab. 2 there are white wholes (lack of lines)

6. In the conclusions (line 328-333) the Authors should underline that all sentences are related to patients with PC.
